# Olfactory bulb-cortex oscillations encode perceived odor intensity rather than concentration

Frans Nordén [1]☯*, Irene Zanettin[1]☯, Mikael Lundqvist[1], Artin Arshamian[1‡],
Johan N. Lundström[1,2,3‡]*

1 Department of Clinical Neuroscience, Karolinska Institutet, Stockholm, Sweden, 2 Department of Otorhinolaryngology, Karolinska University Hospital, Stockholm, Sweden, 3 Monell Chemical Senses Center, Philadelphia, Pennsylvania, United States of America

☯ These authors share first authorship on this work.
‡ These authors share senior authorship on this work.
* Frans.Norden@ki.se (FN); Johan.Lundstrom@ki.se (JNL)

## Abstract

Perceived stimulus intensity is a core feature of sensory experience, yet how it emerges in the human olfactory system remains unknown. Here, we demonstrate that oscillatory dynamics in the human olfactory bulb (OB) and piriform cortex (PC) primarily encode subjective perceived intensity rather than physical concentration. Using noninvasive electrobulbogram recordings, we show that early gamma-band activity in the OB reflects bottom-up transmission of perceived intensity to the PC, which in turn sends top-down beta-band feedback that modulates OB activity via phase–amplitude coupling and transient beta bursts. This bidirectional communication supports a dynamic updating mechanism that maintains perceptual constancy across varying environmental odor concentrations. Our findings reveal a previously uncharacterized oscillatory framework for intensity coding in the human olfactory system, highlighting the primacy of perception over stimulus properties and offering a mechanistic basis for predictive processing in early sensory circuits.

## Introduction

A core function of all sensory systems is to determine the intensity of a stimulus. This basic perceptual processing, where stimulus quantity is transformed into perceived intensity, enables us to differentiate a touch from a punch, a whisper from a scream, a candle from the sun, and a whiff from a stench. Today, we have a detailed mechanistic understanding of how the human auditory and visual systems encode both physical stimulus properties (e.g., volume and luminance) and their perceived perceptual counterparts (e.g., loudness and brightness). This is not the case for the

**Data availability statement:** All anonymized data and scripts required to reproduce the results are available at https://doi.org/10.17605/OSF.IO/T7HJ2.

**Funding:** This work was supported by the Knut and Alice Wallenberg Foundation (KAW 2018.0152; https://kaw.wallenberg.org/; awarded to JNL), the D2Smell ERC Synergy award (101118977; https://erc.europa.eu/; awarded to JNL), and the Swedish Research Council (2024-01605_VR; https://www.vr.se/; awarded to AA). The use of the MR facility at Stockholm University (SUBIC) was made possible by an institutional grant to SUBIC (SU FV-5.1.2-1035-15; https://www.su.se/english/divisions/subic---stockholm-university-brain-imaging-centre). The funders had no role in study design, data collection and analysis, decision to publish, or preparation of the manuscript.

**Competing interests:** The authors have declared that no competing interests exist.

**Abbreviations:** AUC, area under the curve; EBG, electrobulbogram; ITI, inter-trial interval; MI, modulation index; OB, olfactory bulb; PAC, phase–amplitude coupling; PC, piriform cortex; PCC, posterior cingulate cortex; PTR-TOF, proton transfer reaction time of flight; SVM, Support Vector Machine; WCM, weighted cluster mass.

human olfactory system, where nearly all our knowledge of how the brain processes odor intensity comes from animal studies. These animal studies primarily focus on stimulus concentration, as it is difficult to obtain subjective, trial-by-trial ratings of perceived odor intensity from animal models.

The lack of trial-specific information presents a challenge, given that perceived odor intensity is not a direct reflection of the physical concentration of odorant molecules. Rather, it represents a sophisticated perceptual construct in which the brain's early olfactory structures transform the raw chemical input received from odor receptors into the subjective experience of odor intensity. While stimulus concentration serves as the initial sensory signal, perceived intensity exhibits an imperfect logarithmic relationship with concentration [1].

In nonhuman mammals, it is thought that encoding of odor concentration occurs through precise temporal patterns of mitral and tufted cell spiking in the olfactory bulb (OB), where increasing concentration leads to earlier and more synchronized responses [2,3]. These temporal patterns are believed to be further processed in the piriform cortex (PC), where recurrent network dynamics amplify early OB inputs and suppress later activity, forming ensemble codes for concentration [4]. Critically, these mechanisms are embedded in oscillatory neural dynamics with the amplitude and frequency of OB and PC oscillations varying systematically with odor concentration and volatility [5–7]. Together, these studies have laid the foundation for a temporal and oscillatory framework of odor intensity processing; however, this framework remains untested in humans.

No study has yet examined how the human OB processes odor intensity and only a few studies have explored how the human brain more broadly represents odor intensity. Electrophysiological and imaging studies have revealed that neural responses obtained from the scalp or PC are more tightly linked to the subjective perception of intensity than the mere chemical concentration of odorants [8,9]. However, the mechanisms by which the initial nodes within the human olfactory system communicate intensity-related information are still unknown. This knowledge gap is largely due to the dearth of noninvasive methods for recording OB and PC oscillatory responses outside of surgical settings. Recent technological developments have, however, enabled in vivo, noninvasive, and simultaneous electrophysiological assessment of the OB and PC functions in humans. Specifically, we recently developed and validated the electrobulbogram (EBG), an EEG-based method for assessing oscillatory activity of the human OB and PC [10,11].

In the present study, we used the electrobulbogram to investigate how the OB and the PC, two primary nodes of the olfactory system, process and communicate odor intensity at their earliest stages (Fig 1E). We focused on neural oscillations in response to two key aspects: odor concentration and perceived odor intensity (Fig 1A–1D). Given that oscillatory bursts observed in electrical fields have been proposed as proxies for population spiking or activity motifs [12] and previous animal studies have largely focused on spiking activity in relation to intensity, we also assessed burst activity in relation to both concentration and perceived intensity to enable more direct comparisons with the existing literature.

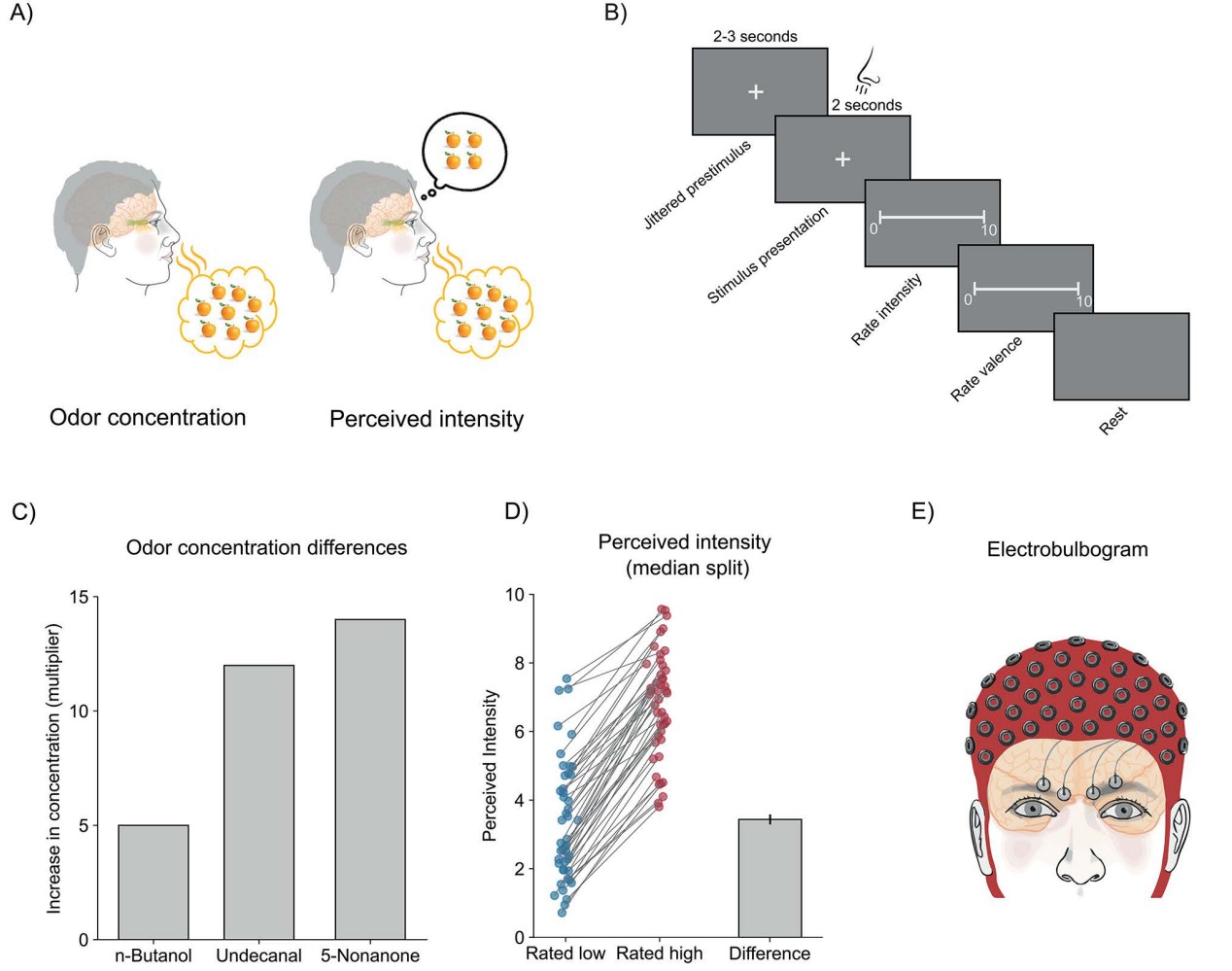

**Fig 1. Stimuli characterization, trial overview, and odor stimuli. A)** Odor concentration is a physical property (number of odorant molecules), whereas perceived intensity reflects the participant's subjective experience. **B)** Single-trial structure. After a jittered pre-stimulus period, odor is delivered at inhalation onset and remains on for 2 s. Immediately after odor offset, participants rate perceived intensity and valence on a 0–10 visual analog scale. The response and subsequent rest period yield a minimum inter-trial interval of 14 s. **C)** Differences between the Low and High concentration conditions for each odor, expressed as concentration multiplier. **D)** Rated perceived intensity based on the median split of all individual trials. Each dot represents the mean rating of a participant with their ratings in both conditions connected with line-marking. The bar graph shows the mean perceived difference between conditions; blue denotes Low, red denotes High, and the error bar represents SEM. Data supporting this figure are available at https://doi.org/10.17605/OSF.IO/T7HJ2. **E)** Method to record signals from the OB and PC.

## Results

### Perceived odor intensity, but not odor concentration, linearly relates to gamma and beta amplitude in the OB and PC

We first determined whether odor concentration or perceived odor intensity is related to frequency amplitudes by assessing each time-frequency point in the power spectrum with a linear mixed-effects model. To control for individual differences in rating behavior, we included Participant as a random intercept, and to control for the influence of perceived odor valence and sniff size, we included perceived valence and the sniff parameters amplitude and AUC as fixed effects.

Statistical significance was evaluated using a Monte–Carlo test with 1,000 permutations and weighted cluster-mass correction.

When examining odor concentration, we found surprisingly little evidence of concentration-related modulation in the time–frequency spectrum. Despite a more than average 10-fold difference in stimuli concentration between weak and strong concentrations (Fig 1C), only small, scattered time–frequency regions in the OB and PC showed any relationship between concentration and power amplitude (Fig 2A, 2B), and none survived cluster correction.

In contrast, perceived odor intensity revealed robust cluster-corrected effects in both neural regions when continuous intensity ratings were modeled as a fixed effect. In the OB, beta-band (12–25 Hz) power was significantly related to intensity ratings from 600 to 1,600 ms after odor onset ($t = 3.51$, $p = .001$, CI = [.0081, .0101]; Fig 2C). A similar beta-band effect in the PC occurred slightly later, from 850 to 2000 ms in the 15–30 Hz range ($t = 3.03$, $p = .004$, CI = [.0142, .0222]; Fig 2D). Both regions showed intermittent beta-band peaks as early as ~200 ms after odor onset; however, these early peaks did not survive cluster correction. Around 1 s into the trial, the association in the PC ($t = 4.65$, $p = 3.36 \times 10^{-6}$, CI = [.0757, .186]) was comparable to the ones in the OB ($t = 4.08$, $p = 4.49 \times 10^{-5}$, CI = [.0726, .210]).

These effects were also visible in the raw power spectra (S3, S4 Figs). When trials were divided by perceived intensity using a median split, one significant cluster was observed in the PC in the gamma band (~30 Hz, 1,500–1800 ms; $t = 3.7$, $p = .018$, CI = [.0064, .0116]). Similarly, dividing trials by low versus high odor concentration revealed a significant cluster in the beta band (~25 Hz, 800–1,200 ms; $t = 3.5$, $p = .049$, CI = [.0094, .0176]) overlapping with the time window identified in the mixed-effects model for perceived intensity.

Gamma-band results from the mixed-effect models showed a similar pattern. In the OB, gamma power (70–100 Hz) was significantly related to perceived intensity between ~1,100 and 1800 ms, and a later cluster emerged in the PC (80–100 Hz) from ~1,400 to 2000 ms. Associations were comparable in the PC ($t = 2.37$, $p = .022$, CI = [.020, .064]) and the OB ($t = 2.41$, p = .02, CI = [.021, .061]). As in the beta band, early OB gamma effects (~200 ms; 70–80 Hz) were present but did not survive cluster correction. The activity was also visible in the raw power spectrum, both for perceived intensity and concentration.

To account for the rapid dynamics of olfactory processing, we repeated the analyses using a restricted time window of the first 500 ms following odor onset. This analysis did not reveal any significant cluster-corrected effects for either perceived intensity or odor concentration.

To assess whether the observed effects of perceived intensity on OB and PC power reflected a cortex-wide phenomenon, we performed the same analysis in two additional control regions: a primary sensory region (primary visual cortex, V1) and a higher-order region implicated in global arousal and attentional processes (posterior cingulate cortex, PCC). Using the identical cluster-based permutation framework, no significant intensity-related clusters were observed in either control region (S6 Fig). These findings suggest that the reported effects are not ubiquitous across the cortex and instead appear relatively specific to the olfactory ROIs.

Animal studies have shown that the OB encodes odor concentration [5–7]. Thus, the lack of clear concentration effects in our time–frequency analysis was unexpected. To test whether concentration effects might be present outside the time-frequency domain, we performed a PCA on raw OB signals for high versus low concentration. The trajectories diverged early in the odor period (0–2000 ms) before converging later (S3 Fig), suggesting concentration-dependent processing that is not captured by time-frequency amplitudes.

We obtained a broad range of intensity percepts for nearly all participants. To verify that this reflected perception rather than idiosyncratic rating behavior, we utilized the established fact that there is a weak but linear relationship between perceived intensity and the sniff response. Subjective ratings correlated positively with sniff AUC ($t = 3.79$, $p = .00015$), whereas physical concentration (High/Low) did not ($t = .19$, $p = .85$). Sniff parameters were included as covariates of no interest in all other models.

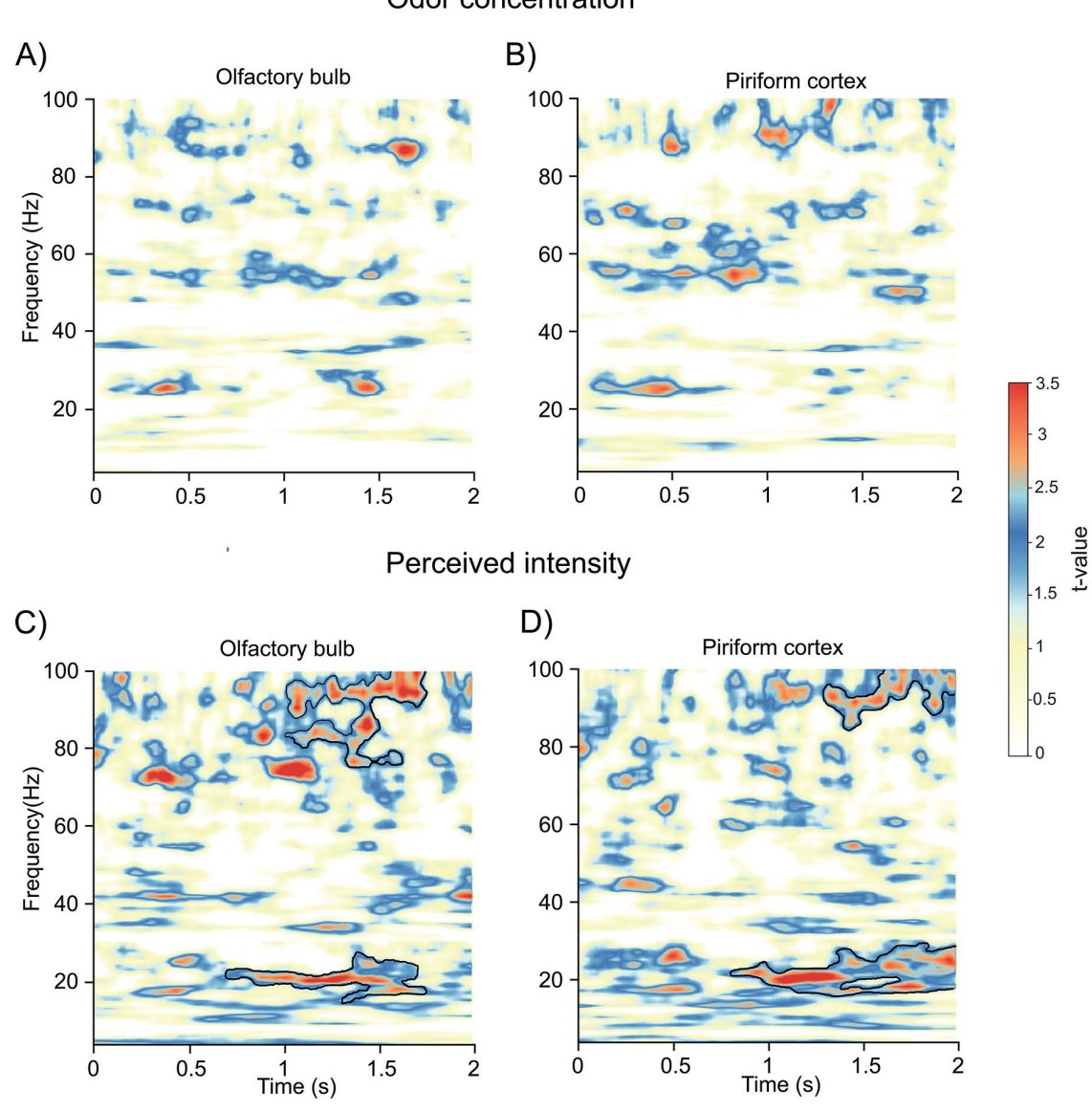

**Fig 2. Gamma and beta power are predicted by perceived intensity, but not actual odor concentration, in OB and PC. A)** Linear mixed-effects model applied to the OB time–frequency spectrum testing whether odor concentration (High vs. Low) predicts power amplitude. Subject was modeled as a random intercept, and perceived valence and sniff parameters were included as fixed effects. Significant clusters after weighted cluster-mass correction are outlined in black. **B)** Same as A, but for PC. **C)** Same model as in A but with perceived odor intensity instead of odor concentration, revealing significant beta- and gamma-band clusters associated with perceived intensity in the OB. **D)** Same as C but applied to the PC, revealing significant beta- and gamma-band clusters associated with perceived odor intensity in the PC. Data supporting this figure are available at https://doi.org/10.17605/OSF. IO/T7HJ2.

Finally, because our primary concentration analysis used a dichotomized variable (High/Low), we examined whether the lack of variance might obscure effects. We transformed the six odor conditions into a continuous "effective concentration" metric (vapor pressure × dilution) and reran all time-frequency analyses (S6A, S6B Fig). We also quantified absolute concentration values, measured at the olfactometer outlet (within the experiment, located about 2 cm into the nose) using

proton transfer reaction time of flight mass spectroscopy (PTR-TOF). These analyses replicated the original finding: no cluster-corrected effects of effective concentration on time–frequency amplitude (S7A, S7B Fig).

## Perceived odor intensity is reciprocally communicated between the OB and PC in the gamma and beta band

We next asked how odor concentration and perceived odor intensity are communicated between the OB and PC. To address this, we first computed the coherence spectrum between the two regions and subsequently applied separate SVM classifiers for binary decoding of either concentration (high/low) or perceived intensity (high/low, median split). The classifier could not decode concentration in any time–frequency region (Fig 3A). In contrast, it successfully decoded perceived intensity from several time–frequency windows (Fig 3B).

The first significant decoding window occurred early, ~200 ms after odor onset, in the higher gamma band (70–90 Hz, $t = 2.56$, $p = .014$, CI = [.0068, .0212]). As shown in the confusion matrix (Fig 3E–3I), the classifier performed better at identifying low perceived intensity (76%) than high (67%). This early gamma effect overlaps temporally with the

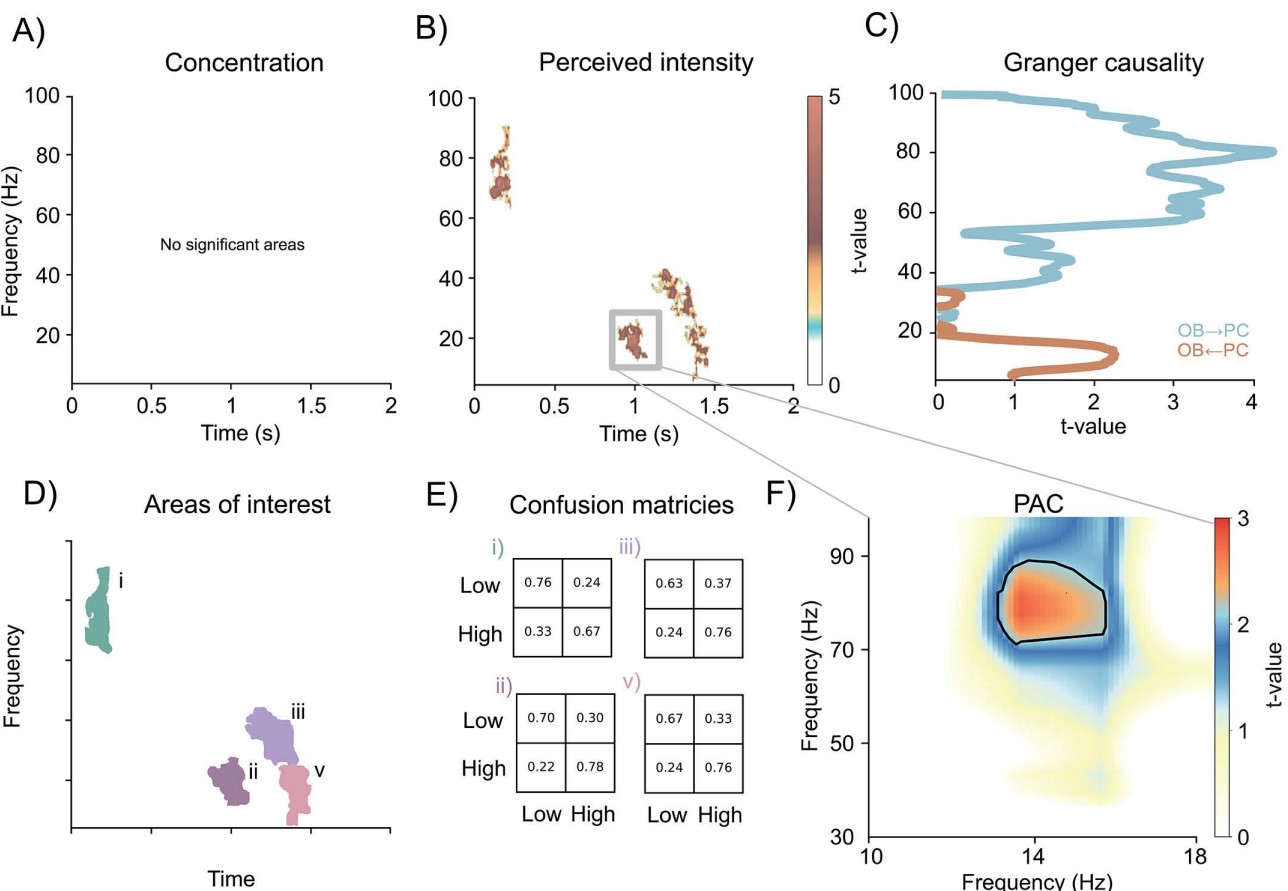

**Fig 3. Perceived odor intensity is communicated between the OB and PC. A)** Classification of high vs. low odor concentration in the coherence spectrogram revealed no significant time–frequency regions. **B)** In contrast, classification of high vs. low perceived intensity showed significant decoding accuracy in both the gamma and beta bands. **C)** Granger causality demonstrated information flow from OB→PC in the gamma band, and from PC→OB in the beta band, indicating that the beta-band region identified in B reflects top-down signaling from PC to OB. **D)** Time–frequency regions of interest identified in B. **E)** Confusion matrices for the four significant time–frequency clusters shown in B. **F)** Phase–amplitude coupling (PAC) within the 950–1,100 ms window showed beta-phase modulation of gamma-amplitude in the OB, consistent with PC-to-OB feedback at the time when perceived intensity is communicated back to the OB. Data supporting this figure are available at https://doi.org/10.17605/OSF.IO/T7HJ2.

gamma-band activity observed in the linear mixed-effects analysis of the OB and PC power spectra (Fig 2C, 2D). Although those early OB/PC gamma effects did not survive cluster correction they were significant on the voxel level. Together, the converging evidence suggests that both OB and PC encode perceived intensity at this early stage.

A second decoding window was observed around 1 s after odor onset in the beta band (12–25 Hz; $t=2.56$, $p=.014$, CI = [.0068, .0212]). This time period coincides with significant beta clusters identified in the mixed-effects analyses for both OB and PC. Classification accuracy in this region was highest overall (74%), with better performance for high perceived intensity (78%) than low (70%).

A third cluster spanned 1,200–1,500 ms and extended from beta into low gamma frequencies (10–38 Hz; $t=3.03$, $p=.004$, CI = [.0001, .0079]). Because this cluster covered both frequency ranges, we examined classification performance separately for beta and gamma. In both bands, high perceived intensity was decoded more accurately (76%) than low (67% in beta; 63% in gamma).

Odors were selected to be neutral in valence and displayed little variability (S5A Fig). Moreover, perceived valence was included in analyses as a covariate of no interest. Still, to ensure that coherence-based decoding was not driven by valence, we conducted a parallel classification of perceived valence (median-split procedure on trials and participants). As shown in S5B Fig, there was no overlap between perceived valence and perceived intensity in the coherence spectrum.

To assess the direction of information flow, we performed frequency-resolved Granger causality, contrasting OB→PC and PC→OB to remove shared variance. High gamma band connectivity reflected information transfer from OB to PC ($t=5.04$, $p=8.3e{-}6$, CI = [0.2275, 0.5313]; Fig 3C). In contrast, lower frequencies reflected information transfer from PC to OB ($t=2.46$, $p=.0134$, CI = [0.0833, 0.8417]). Thus, the beta-band activity observed around ~1 s in the coherence spectrum likely corresponds to top-down communication from PC to OB.

Guided by this hypothesis, we performed a PAC-analysis in this window and found that beta phase (12–16 Hz) modulated gamma amplitude (70–90 Hz) as a function of perceived intensity ($t=2.35$, $p=.023$, CI = [.0137, .0323]; Fig 3F). Granger causality also indicated that the beta-band region observed around ~1,400 ms falls within a window of significant PC→OB information transfer, consistent with continuous updating of OB representations by the PC.

## PC beta and OB gamma bursts indicate information transfer

Analyzing intermittent burst activity provides additional insights beyond those offered by traditional wave-based analyses because it considers that oscillatory activity is intermittent and vary from trial to trial. This aligns with the view that cognition and executive control are supported by rapid transitions between discrete, transient neural states, rather than by slow, continuous dynamics [13]. Furthermore, oscillatory bursts are closely linked to population spiking, making them a useful proxy for ensemble activity. Thus, to further investigate the relationship between odor intensity processing, cognition, and neural oscillations, we analyzed single-trial burst activity in the beta and gamma bands during odor presentation, examining its association with both odor concentration and perceived intensity. Burst counts within each frequency band were extracted for every trial, and their association with concentration and perceived intensity was evaluated using linear mixed-effects models, with statistical significance assessed via a Monte–Carlo permutation test (1,000 permutations) using weighted cluster-mass correction.

When relating odor concentration to burst activity, we found no beta or gamma effects in the OB that survived cluster correction. We did find, however, at an uncorrected level ($p<.05$) that beta burst rate was associated with concentration from 1,500 to −1800 ms into the trial ($t=2.80$, $p=.0051$, CI = [7.70e−5, 5.85e−4]). A similar, likewise uncorrected pattern was observed in the PC at the same time ($t=2.87$, $p=.004$, CI = [1.50e−4, 7.94e−4]).

In contrast, perceived intensity produced robust, cluster-corrected effects. In the OB, gamma bursts were significantly related to perceived intensity at two distinct time points. The first occurred 800–1,100 ms after odor onset ($t=2.88$, $p=.006$, CI = [.0012, .0108]). Interestingly, after this event, the OB gamma burst activity then became anti-correlated to perceived odor intensity, suggesting suppression of high-frequency bursts by intensity. Later, burst activity in the gamma

band was again positively related to odor intensity ratings between the timepoints 1,350–1,450 ms ($t = 2.53$, $p = .015$, CI = [.0075, .0225]).

In the PC, perceived intensity was associated with significant beta burst activity in a time window that closely matched the initial OB gamma burst effect (800–1100ms; $t = 2.28$, $p = .027$, CI = [.0170, .0370]). Critically, this is the same period in which the coherence and Granger analyses indicated top-down information transfer from PC to OB in the beta band, and the same interval in which phase–amplitude coupling linked PC beta phase to OB gamma amplitude.

## Discussion

We demonstrate here that oscillatory activity within the OB and PC is clearly associated with perceived odor intensity, but only weakly with physical odor concentration. Specifically, we show that perceived odor intensity information, after initial processing in the OB within the gamma band, is transmitted as a bottom-up signal from the OB to the PC via gamma-band oscillations and is subsequently modulated through top-down feedback from the PC to the OB via beta-band oscillations. This top-down modulation updates the internal state of OB gamma activity through phase-amplitude coupling, driven by an underlying burst-like dynamic. No similar patterns were found for physical odor concentration, suggesting that oscillatory activity in the human OB and PC primarily reflects perceptual processing rather than stimulus concentration. This constitutes the first direct evidence in humans that subjective odor intensity, not physical concentration, is encoded through coordinated oscillatory dynamics between the OB and PC. These findings challenge the traditional view that early olfactory processing reflects stimulus concentration and instead reveal that perception is already manifested at the earliest hierarchical stages of the human olfactory system.

We found a strong link between OB processing and the subjective perception of odor intensity, suggesting that by the time oscillatory population activity is measured (hundreds of milliseconds post-stimulus), it reflects a perceptually transformed representation. This view is supported by rodent studies showing that the OB can encode internal variables corresponding to perceived intensity rather than just reporting concentration linearly [3], and that intensity coding can involve structured population dynamics [14].

At the same time, animal research has also shown that the OB processes odor concentration [2,3], so the lack of a clear link in our data between stimulus concentration and neural processing was therefore unexpected. However, this absence of concentration effects cannot be attributed to nonlinear liquid–vapor relationships [15]. We directly measured vapor-phase concentrations using PTR-TOF mass spectrometry and repeated all analyses using empirically derived concentration values. These analyses yielded results indistinguishable from those based on nominal dilution levels. Although model-free PCA of raw OB signals revealed early divergence between concentrations, these effects were not reflected in oscillatory amplitude measures. Thus, while concentration-dependent processing may occur at early sensory stages, it does not appear to dominate the coordinated oscillatory dynamics measured here.

Notably, an additional cluster-based analysis revealed a small late concentration-related effect in the beta range that overlapped temporally with the intensity-related activity. However, this effect was not observed in the linear mixed-effects model and was less consistent across analytical approaches. The temporal overlap with intensity-related dynamics suggests that such activity may reflect shared or interacting processes rather than a distinct encoding of physical concentration. Thus, while some late concentration-related signatures may be present, they appear weaker and less robust than the effects associated with perceived intensity.

These findings align with a broader view of intensity coding in animal models, where concentration is not represented simply through increased firing rates, but through temporal and population-level dynamics. In the rodent piriform cortex, overall firing rates remain relatively stable across concentrations for a given odor identity, whereas concentration is reflected in latency shifts and evolving ensemble patterns [2]. Specifically, the piriform cortex uses a dual-component code where an initial subset of neurons responds rapidly and largely independent of concentration (carrying odor identity), while a later subset responds with latency shifts as concentration changes, thus conveying odor intensity [2]. Such temporal

coding principles have also been described in insects and fish [7,16], suggesting that dynamic timing, rather than rate alone, underlies magnitude representations across species.

Our findings extend this framework to humans, demonstrating that subjective intensity, rather than physical concentration, is reflected in early human OB–PC communication via gamma-band oscillations around 200 ms post-stimulus. This timing coincides with the time window when concentration becomes decodable in the rodent PC. Similarly, prior work using the electrobulbogram method has shown that odor identity can be decoded from OB-PC gamma connectivity at earlier latencies (100 ms) [11]. Together, these results suggest that the human olfactory system employs a temporally staged sequence, where identity and intensity are communicated along distinct OB–PC gamma-band channels in rapid succession. Crucially, the later beta-band feedback (1 s post-stimulus) from PC to OB, strongly tied to perceived intensity, may act to refine or reinforce this perceptual representation. This interpretation is consistent with the known role of the piriform cortex in integrating sensory input with contextual and cognitive signals, and with theories that beta oscillations coordinate top-down modulation in olfactory circuits [14,17,18]. Together, these findings support a temporally resolved model in which subjective odor intensity emerges through iterative feedforward and feedback interactions within the human OB–PC loop.

Our observation that gamma oscillations mediate bottom-up transfer while beta oscillations mediate top-down feedback resonates strongly with theoretical and experimental work in animal models. It has long been proposed that fast gamma oscillations primarily reflect local circuit processing within the OB and the initial feedforward relay of olfactory information, whereas slower beta oscillations engage a larger loop including the olfactory cortex and potentially higher-order areas [17,19]. For example, OB beta oscillations are typically weak or absent under anesthesia or when cortical feedback is disrupted, while gamma oscillations persist, indicating that beta rhythms critically depend on an intact OB-PC loop [19,20]. Our data, obtained in humans, demonstrates significant Granger-causal influences from OB to PC in the high gamma band, and conversely from PC to OB in the beta band during intensity processing, supporting this model.

This sequential processing of odor intensity mirrors a two-stage scheme proposed in olfaction and other sensory systems [11,16,18,21,22], wherein an initial feedforward sweep is followed by feedback that fine-tunes the representation, which fits with the timing of odor-evoked gamma and beta oscillations found in human intracranial PC recordings [23]. This OB-PC loop could potentially be viewed through a predictive coding lens [24], where the cortex sends predictions to early sensory areas, and discrepancies (prediction errors) drive updates of internal representations to refine perception [25,26]. The observed gamma/beta cycle could implement such a mechanism, iteratively minimizing the mismatch between expected and actual intensity.

Importantly, the association between gamma and feedforward processing, and beta and feedback processing, should be regarded as a functional heuristic rather than a strict rule. We therefore base our interpretation not on frequency alone, but on the combined evidence of temporal ordering, directionality, behavioral relevance, and consistency with known olfactory circuitry. Within these constraints, the OB–PC dynamics observed here provide converging evidence for recurrent feedforward and feedback interactions underlying subjective odor intensity processing, paralleling communication principles described in other sensory systems [27].

Evidence that the internal representation of the OB is updated is found in single-trial analysis of our data showing intermittent oscillations [13,28,29], and from the observation that the rate and timing of bursts in the beta and gamma bands are systematically related to perceived intensity. Specifically, OB gamma-band bursts showed a biphasic positive relationship with higher perceived intensity (an initial surge at ~0.8–1.1 s, followed by suppression, then a later surge at ~1.35–1.45 s). Crucially, PC beta-band bursts occurred in nearly the same early time window (~0.8–1.1 s) as the OB's initial gamma burst surge. This temporal coincidence, coupled with our finding that beta phase in the PC modulated OB gamma amplitude (PAC) during this period, suggests a sequential interaction: as the PC evaluates intensity (reflected in a PC beta burst), its feedback triggers or "gates" OB gamma bursts, thereby updating the OB's representation in discrete epochs. This aligns with emerging views of cortical function where information is processed in transient "packets" or bursts, facilitating efficient communication and plasticity. Gamma bursts are thought to reflect synchronized spiking in local

networks, while beta bursts often involve more distributed networks [17,29]. Thus, a PC beta burst, possibly influenced by higher-order inputs, could prime piriform ensembles to send temporally precise volleys of activity that manifest as synchronized gamma bursts in the OB, ensuring robust and adaptive population coding of intensity. Such within-sniff updates, occurring over hundreds of milliseconds, are sufficiently fast to influence perception in real time.

Interpreting frequency-specific and directional interactions from noninvasive EEG signals requires care. Granger causality captures directed statistical dependencies between reconstructed signals rather than direct biological causation, and spectral peaks in Fourier-based analyses can reflect transient or nonsinusoidal events rather than sustained oscillations [30–32]. We therefore do not claim to have demonstrated causal neural mechanisms in a strict biological sense. What we do claim is that the pattern of effects, replicable across directed connectivity, phase–amplitude coupling, and single-trial burst dynamics, and consistent with established gamma and beta circuit mechanisms in animal models of olfaction [33–35] is precisely what one would predict from an OB–PC circuit implementing feedforward and feedback processing of odor intensity. The convergence of independent measures, rather than any single analysis, constitutes the evidential basis for our interpretation. Together, these observations support the presence of temporally structured, frequency-specific interactions within the OB–PC circuit that are consistent with known anatomy and task timing.

A critical question is whether effects show anatomical specificity. We therefore repeated the analyses in the primary visual cortex and the posterior cingulate cortex. No comparable intensity-related effects were observed under identical statistical criteria, arguing against a global anticipatory or task-structure account. Combined with the directionality and cross-frequency coupling observed within the OB–PC loop, these findings support the interpretation that late beta activity reflects recurrent refinement of perceived intensity rather than nonspecific expectancy effects. However, this does not exclude the involvement of content-specific expectancy mechanisms. Accordingly, the late beta activity may reflect an interaction between perceptual refinement and task-related processes within the olfactory system. Future studies using passive odor presentation or delayed-response paradigms will be important to disentangle these components.

In summary, the human olfactory bulb and piriform cortex engage in a dynamic, frequency-specific dialogue to represent perceived odor intensity. In this view, even the most fundamental aspect of perception becomes an active, dynamic, and contextually informed process. Initial bottom-up gamma oscillations convey early sensory information, which is subsequently refined by top-down beta-band feedback from the cortex, implemented through PAC and coordinated neural bursts. This framework highlights the primacy of subjective perception in shaping early olfactory processing and suggests that, similar to other sensory systems, the olfactory bulb and piriform cortex rely on iterative network interactions to construct a stable and behaviorally relevant representation of the world.

## Materials and methods

### Ethics statement

Participants were informed about the purpose and procedure of the study and provided written informed consent before participation. The study was approved by the Swedish National Ethical Review Board, Etikprövningsnämnden (EPN: 2017/2332-31/1), and was performed in accordance with the Declaration of Helsinki.

### Participants

A total of 53 individuals participated in the study. Following the exclusion of 7 participants, as described in the preprocessing section, the final sample included 46 participants (mean age 30, SD ± 8.8; 27 were women). Participants with functional anosmia or severe hyposmia were excluded from the study. Olfactory function was assessed using a brief odor identification screening test consisting of five common odors. For each odor, participants were asked to choose the correct label from four options. A minimum of three correct responses was required to meet the inclusion threshold. Participants scoring below this cutoff were excluded to ensure adequate baseline olfactory function.

## Testing procedure

Testing was conducted in a sound-attenuated and well-ventilated recording booth, built for odor testing and designed to ensure no-to-limited background odors. During the testing session, participants wore earplugs and headphones that played low-volume white noise to mask any potential auditory cues from the olfactometer, odor delivery setup, or other external sounds. Event timing and stimulus triggering were implemented using PsychoPy 3 [36]. Because stimulus presentation was triggered by the individual's nasal inspiration, participants were instructed to breathe normally through their nose throughout the experiment without needing to purposely time the odor onset. The experiment consisted of four 15-min blocks with a break between each one (Fig 1D).

Each block consisted of 35 trials, resulting in a total of 140 experimental trials per participant (3 odors with Low/High concentration and Clean air, 20 trials each). Odors were delivered in a randomized sequence, with their presentation evenly balanced across the testing blocks. Following each trial, participants were instructed to rate the intensity and pleasantness of the stimulus on a visual analogue scale containing 100 steps (operationalized as 0.0–10.0). Rating scales were anchored with the text "not perceived/very unpleasant" and "very intense/very pleasant" at opposite ends. The inter-trial interval (ITI) was jittered but set to at least 14 s to limit odor habituation.

## Odor delivery and odor stimuli

Three neutral odors, chosen from different chemical classes, were selected based on extensive in-house testing. Each odor was diluted into two different concentrations, resulting in two subsets of odors (Low and High). Using odors from three distinct chemical classes allowed us to draw conclusions that are more likely to generalize across odor processing and to minimize odor-specific effects. In addition, a no-odor condition, i.e., clean air trials in which the ongoing airflow was replaced with clean air, was presented. All air used in the experiment underwent cleaning stages to remove unrelated odors potentially originating from the compressor system, where the air was passed through both active coal and micro filters. Low-concentration odors were obtained by diluting neat n-Butanol (an alcohol, Fisher Chemicals, CAS 71-36-3), Undecanal (an aldehyde, Sigma-Aldrich, CAS 112-44-7), and 5-Nonanone (a ketone, Sigma-Aldrich, CAS 502-56-7) to 0.8%, 1.4%, and 1.3% volume/volume in diethyl phthalate (99.5% pure, Sigma-Aldrich, CAS 84-66-2), respectively; while high-concentration odors were obtained by diluting the aforementioned odors to 4% (a 5-fold increase in concentration), 17% (a 12-fold increase), and 18% (a 14-fold increase) in diethyl phthalate, respectively (Fig 1B). Concentrations were selected so that odorants within each subset (Low and High) would be iso-intense; as confirmed through pilot studies. Odors were delivered birhinally for 2 seconds per trial using a computer-controlled 8-channel olfactometer and presented using a sniff-triggered design [37]. A thermo-pod (sampling rate of 400 Hz; PowerLab 16/35, ADInstruments, Colorado) with a temperature probe inserted just inside participants' nostrils monitored the breathing pattern based on intranasal temperature. Based on the breathing signal, an individual threshold was set to trigger the olfactometer at the beginning of the inhalation phase to synchronize odor presentation with nasal inspiration and avoid inducing attention-related EEG artifacts. To avoid any potential tactile sensation at the odor onset, a birhinal airflow of 2.7 liters per minute was used for odor delivery and inserted into an ongoing constant airflow of 0.3 liters per minute of clean air. The olfactometer has an onset delay of ~150 ms. This delay, representing the time required for the odor to reach the nasal epithelium, dependent on tubing length and airflow, was verified using a photo-ionization detector (Aurora Scientific, Ontario) prior to the start of the study. Critically, the established odor delivery delay was taken into account in subsequent analysis, meaning that time point zero in all results corresponds to when the odor enters the nostrils.

Perceived odor intensity is imperfectly related to odor concentration [1]. Whereas there tends to be a general agreement between a specific odorant's concentration and how intense it is perceived, the percept can vary between individual trials based on a range of factors, such as intranasal environment, perceived intensity of prior trials, adaptation and habituation, sniff behavior, sampling duration, rating behavior, trigeminal components, attention, among others. This means

that over the course of an experiment, a given concentration of an odor can produce a range of perceived intensities that are a true percept and not solely attributable to rating behavior. Across all trials, the distribution of rated perceived intensity for the six odorant conditions produced intensity percepts spanning nearly the full rating range with a slight overrepresentation around predicted Low/High concentrations (S1 Fig). To assess neural responses to perceived odor intensity, as outlined below, we used trial-by-trial ratings. The average min-max range of ratings across participants was $8.28 \pm 1.72$ (S2 Fig), demonstrating a good range of ratings. For our support vector machine classification analyses, which require a binary factor, we performed a median split for each individual to create low and high-rated intensity trials, independent of the stimulus. The low-rated intensity trials were on average rated at 3.36 ($\pm$ 2.14 SD), and the high-rated intensity trials were on average rated as 6.80 ($\pm$ 1.86 SD), $t(45) = 23.33$, $p < 8.93\mathrm{e}{-27}$, corresponding to an average difference of 102% between low and high (Fig 1C).

## EEG and EBG recordings

Continuous neural activity was recorded at a frequency of 512 Hz using 64 EEG and 4 EBG active electrodes (ActiveTwo, BioSemi, Amsterdam, the Netherlands) (Fig 1E). EEG electrodes were placed following the international 10/20 system and EBG electrodes were placed on the forehead slightly above the eyebrows. The position of each electrode was digitized in stereotactic space using a neuronavigation system (Brain-Sight, Rogue Research, Montreal, Canada). The digitization protocol involved localizing fiducial landmarks (nasion, left and right preauricular points) and the central point of each EEG/EBG electrode. The identified landmarks were used to map each electrode to standard MNI space. In subsequent analysis, the recorded electrode coordinates were used in the eLORETA algorithm to determine the sources of neural signals, described below.

During recording, the signal was high-pass filtered at 0.10 Hz and low-pass filtered at 100 Hz in the ActiView software (BioSemi, Amsterdam, the Netherlands). Prior to the actual EEG/EBG recording, the impedance of each electrode was visually inspected, and electrodes with an offset greater than 40 mV were manually adjusted until they met a value below the accepted threshold.

## Signal processing and analysis

**Preprocessing.** Prior to preprocessing, five subjects were removed due to too little variability in rated perceived intensity in that they rated all odors nearly identically. For the remaining 48 participants, data were epoched into 5,000 ms segments, from 1,000 ms pre-stimulus to 4,000 ms post-stimulus, and re-referenced to the average activity of all electrodes. Data were then high-pass filtered at 1 Hz and a notch filter at 50 Hz was applied to reduce power line interference. Faulty electrodes were detected through visual inspection and corrected using spline interpolation. Trials with large muscle movements were detected and removed by extracting z-scored Hilbert transform amplitude values with a threshold of 7. Furthermore, eye blinks were removed using Independent Component Analysis with the InfoMax algorithm. Two participants with less than half of the trials left after this step were removed from further analysis. After preprocessing, the remaining final 46 participants had an average of $123 \pm 6.84$ trials remaining for final analyses.

**Source time-course reconstruction.** The source time-course for the OB and PC dipoles was reconstructed using a conduction model based on structural MRI of the participant's head, which has been shown to yield the strongest signal from the olfactory bulb [38]. The structural MRI was segmented into five materials, namely cerebrospinal fluid, gray matter, white matter, scalp, and skull with conductivities [1.79, 0.33, 0.14, 0.43, and 0.01] [39]. The digitized electrode positions were co-registered to the head model and the forward problem was solved using the Finite Element Method through SimBio [40].

A cortex model built on icosahedral resolution 7 generated in Freesurfer based on individual structural MRI was used as a source model. This source model was then transformed into MNI stereotactic space to ensure that identical regions were used for all participants. This source model was used to attain the source activity through solving the inverse

problem with eLORETA with a regularization parameter set to 10%. Singular Value Decomposition was subsequently used to project the source activity along the principal axis. The analysis was then constrained to four ROIs where the dipoles correspond to left $(x-4, y+40, z-30)$ and right OB $(x+4, y+40, z-30)$, determined based on T2 weighted images, as well as left $(x-22, y+0, z-14)$ and right PC $(x+22, y+2, z-12)$ [41]. Additional control regions were included in the analysis: primary visual cortex (V1; $(x\pm8, y-76, z+10)$, corresponding to Brodmann area 17 as defined in probabilistic cytoarchitectonic maps in MNI space [42], and posterior cingulate cortex (PCC; $(x\pm0, y-53, z+26)$, corresponding to BA23/31 based on established cytoarchitectonic mappings in MNI space [43]. The source reconstruction was performed using Fieldtrip toolbox 2023 [44] within Matlab 2023a.

**Linear modelling of the intensity percept.** To evaluate how odor concentration and intensity is processed within the OB and PC, frequency spectra were generated using the Superlet method [45] with width of 8 and an adaptive order based on frequency bands. This approach was used to achieve high resolution in both time and frequency. To assess the relationship between power amplitude and odor concentration or perceived intensity, a linear mixed-effects model was applied to each time-frequency point in the spectrum. The spectral power was used as the outcome variable, while perceived intensity or concentration, together with perceived valence, sniff amplitude, and area under the curve (AUC) of the sniff signal, were used as fixed effects and perceived intensity and valence were treated as continuous variables. We included perceived valence and sniff parameters to ensure that neural effects were not explained by variation in the sniff or odor valence. A random intercept was defined on the subject level according to the formula (power ~ intensity + valence + breathing amplitude + breathing AUC + (1|subject)) or (power ~ concentration + valence + breathing amplitude + breathing AUC + (1|subject)). To assure normality of the residual distribution, an inverse rank transformation was applied to the power spectrum and normality was evaluated with a Shapiro–Wilk test. To consider the problem of multiple comparisons, a cluster-based permutation test with 1,000 permutations was implemented using the weighted cluster mass algorithm (WCM) [46].

**Source connectivity.** When investigating connectivity between neuronal populations, both functional and effective connectivity approaches can be taken. While amplitude and phase relationships are evaluated through functional connectivity [47,48], effective connectivity instead determines the predictive relationship between the two [49]. To understand when two neuronal populations are connected, and which one of them causally influences the other, both approaches are informative.

Functional connectivity was evaluated through the coherence spectrum, allowing us to evaluate whether there was a linear transfer of information between the OB and PC. OB-PC coherence spectra were calculated through Fourier transforming the source reconstructed time-course with the previously mentioned Superlet method. Effective connectivity was evaluated through spectral Granger causality. Granger causality assesses whether the future of a time series (X) can be predicted by past values of X alone or if it is more accurately predicted by past values in the alternative time series (Y) [50]. Spectral Granger causality builds upon the same concept, but the assessment is here performed in Fourier space. To this end, we applied a multi-tapered fast Fourier transform with a DPSS taper-based frequency smoothing of 3 Hz. The Fourier transform was applied to the whole stimulus period and averaged over both hemispheres to increase statistical power. Statistical significance was determined on a group level by applying a two-tailed Student $t$ test.

**Support Vector Machine classification.** A Support Vector Machine (SVM) classifier was applied to the trial-averaged OB-PC coherence spectrogram to determine whether any significant transfer of information related to either perceived intensity or odorant concentration could be decoded. The entire coherence spectrogram was analyzed in a binned searchlight manner, in which the bins measured 12 Hz and 100 ms. Bins with less than 10 neighbors were excluded from further analysis. A leave-one-out scheme was applied to the data in which each participant was left out once per squared area. The accuracy on the group level was compared with a distribution of 1,000 classification results in which the labels were shuffled. A Monte-Carlo simulated one-sided nonparametric $t$ test was used to evaluate significance and significant clusters were evaluated with the WCM algorithm [46].

**PLOS Biology**

**Phase-amplitude coupling.** The integration of neural activity in the brain is believed to be facilitated by a phenomenon wherein the phase of lower-frequency electrophysiological oscillations modulates the amplitude of higher-frequency oscillations, commonly referred to as phase-amplitude coupling (PAC). Our previous studies have demonstrated that perceptual information is communicated back to the OB from the PC in the beta band and subsequently processed within the gamma band in the OB [18]. Therefore, to evaluate information transfer in between frequency bands, we assessed PAC based on the tort algorithm [51] between the beta and gamma bands in the timepoints where our analyses in the coherence spectrum demonstrated information transfer from the PC to OB. A time window of 150 ms was used for the PAC-analysis, based on the duration of the beta activity in the coherence spectrum. The trial level modulation index (MI) derived from the PAC was transformed with an inverse normal rank transformation and then included as the response variable in a linear mixed-effect model with perceived intensity, perceived valence, breathing amplitude, and AUC of the breathing signal as fixed effects and a random intercept based on participant. Normality of the residual distributions was assessed with a Shapiro–Wilk test, and the resulting clusters were corrected for multiple comparisons with a 1,000 permutations Monte–Carlo test with WCM cluster correction.

**Burst analysis.** To investigate oscillatory bursts in response to odor stimulation, we analyzed time-frequency representations of the neural activity recorded during odor perception in which bursts were identified as oscillatory activity in the power spectrum exceeding 1 standard deviation above 10-trial mean and lasting for at least three cycles of the band's average frequency [52]. Because of their intermittent nature, burst events were determined at the single-trial level and extracted for a window of 2 s following stimulus onset. For each participant, we first calculated the burst rate separately for OB and PC and the two frequency bands of interest (beta [12–30 Hz], and gamma [30–100 Hz]). Burst rates were calculated across trials and in some analyses combined across participants, returning a grand average burst rate, per frequency band and region. For the main analysis, we kept single-trial data in order to examine the relationship between burst activity and behavioral parameters (perceived intensity, valence, and odor concentration). We subsequently applied linear mixed-effects models using separate models for beta and gamma bursts in OB and PC. Participants were included as random effects to account for inter-individual variability, and t-statistics for the fixed-effect predictors were used to determine their significance in influencing burst dynamics. To ensure that the residuals of the linear models fulfilled the requirement of normality, a Box-Cox transformation was performed on the response variable.

## Supporting information

**S1 Fig.** Histograms of low (A) and high (B) concentration odor intensity ratings. Solid line represents the density curve (kernel bandwidth = 0.1). This demonstrates that ratings were evenly distributed across the scale with an overrepresentation of ratings at two scale points, in-line with the two odor concentrations used. Data supporting this figure are available at https://doi.org/10.17605/OSF.IO/T7HJ2.
(EPS)

**S2 Fig.** Odor intensity ratings for each individual trial, highlighting in blue low concentration odors and in red high concentration odors. Trials are sorted diagonally within each participant according to participant number (assigned in the order of entering the study). This demonstrates that participants experienced odor intensities across the scale and that there is no systematic skewness of ratings across study time. Data supporting this figure are available at https://doi.org/10.17605/OSF.IO/T7HJ2.
(EPS)

**S3 Fig.** Power spectrum of low and high concentrations for OB and PC, respectively. Data supporting this figure are available at https://doi.org/10.17605/OSF.IO/T7HJ2.
(EPS)

**S4 Fig.** Power spectrum of perceived intensity for OB and PC, respectively. Data supporting this figure are available at https://doi.org/10.17605/OSF.IO/T7HJ2.
(EPS)

**S5 Fig.** Trajectory of the power spectrum in the first three principal components, extracted from the signal source data from the olfactory bulb.These results suggest that there is variance in the odor concentration data that we do not capture with our linear statistical models. Green dots represent time points where we could distinguish between high and low concentrations. Data supporting this figure are available at https://doi.org/10.17605/OSF.IO/T7HJ2.
(EPS)

**S6 Fig.** Similar to Fig 2, but with effective concentration, here denoted as the dilution multiplied by the odor's vapor pressure.There was no significant activity for odor concentration across the time/frequency spectra. This obtained result is similar to what was obtained using the dichotomous division into high and low. Data supporting this figure are available at https://doi.org/10.17605/OSF.IO/T7HJ2.
(EPS)

**S7 Fig.** Power spectrum predicted by absolute concentration for **A**) OB and **B**) PC. Absolute concentration was measured with photon transfer reaction time of flight mass (PTR-TOF) spectroscopy at the outlet of the olfactometer. The values measured for the odors are: N-butanol: (low = 9 ppbV, high 12 ppbV), 5-nonanone (low = 50 ppbV, high = 300 ppbV), undecanal (low = 7 ppbV, high = 23 ppbV). Because nominal liquid dilutions do not necessarily translate linearly into vapor-phase concentrations, particularly at higher concentrations [15], the use of the vapor pressure metric can represent a limitation. Therefore, we measured the vapor-phase concentrations of all odorant stimuli used in the experiment deploying PTR-TOF mass spectrometry. These measurements confirmed that vapor-phase concentrations increased monotonically across the nominal dilution steps, but that the magnitude of the increase in vapor concentration differed across conditions and was not uniformly proportional to nominal liquid dilution factors, consistent with known nonlinear liquid–vapor relationships. Data supporting this figure are available at https://doi.org/10.17605/OSF.IO/T7HJ2.
(EPS)

**S8 Fig.** Power spectrum predicted by odor intensity in **(A)** primary visual cortex V1 and **(B)** posterior cingulate cortex PCC. Data supporting this figure are available at https://doi.org/10.17605/OSF.IO/T7HJ2.
(EPS)

**S9 Fig. Classification of pleasant and unpleasant as a control.A)** Perceived valence did not vary between the two concentrations. High numbers denote pleasant, and low numbers denote unpleasant, perceptual ratings. **B)** None of the areas found overlaps with the areas found in the classification of Intensity. Data supporting this figure are available at https://doi.org/10.17605/OSF.IO/T7HJ2.
(EPS)

## Author contributions

**Conceptualization:** Frans Nordén, Mikael Lundqvist, Artin Arshamian, Johan N. Lundström.

**Data curation:** Frans Nordén, Irene Zanettin.

**Formal analysis:** Frans Nordén, Irene Zanettin.

**Funding acquisition:** Artin Arshamian, Johan N. Lundström.

**Investigation:** Frans Nordén, Irene Zanettin.

**Methodology:** Frans Nordén, Irene Zanettin, Mikael Lundqvist, Artin Arshamian, Johan N. Lundström.

Project administration: Irene Zanettin, Johan N. Lundström.

Resources: Artin Arshamian, Johan N. Lundström.

Software: Johan N. Lundström.

Supervision: Mikael Lundqvist, Artin Arshamian, Johan N. Lundström.

Validation: Frans Nordén, Irene Zanettin.

Visualization: Frans Nordén, Irene Zanettin.

Writing – original draft: Frans Nordén, Irene Zanettin, Artin Arshamian, Johan N. Lundström.

Writing – review & editing: Frans Nordén, Irene Zanettin, Mikael Lundqvist, Artin Arshamian, Johan N. Lundström.

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
