## [Editor Report · Decision Letter 0]

24 Nov 2025

Dear Dr Nordén,

Thank you for submitting your manuscript entitled "Olfactory bulb-cortex oscillations encode perceived odor intensity rather than concentration" for consideration as a Research Article by PLOS Biology.

Your manuscript has now been evaluated by the PLOS Biology editorial staff, as well as by an academic editor with relevant expertise, and I am writing to let you know that we would like to send your submission out for external peer review.

Once your full submission is complete, your paper will undergo a series of checks in preparation for peer review. After your manuscript has passed the checks it will be sent out for review. To provide the metadata for your submission, please Login to Editorial Manager (https://www.editorialmanager.com/pbiology) within two working days, i.e. by Nov 26 2025 11:59PM.

Kind regards,

Taylor

Taylor Hart, PhD,

Associate Editor

PLOS Biology

thart@plos.org

---

## [Decision Letter · Decision Letter 1]

5 Feb 2026

Dear Dr Nordén,

Thank you for your patience while your manuscript "Olfactory bulb-cortex oscillations encode perceived odor intensity rather than concentration" was peer-reviewed at PLOS Biology. It has now been evaluated by the PLOS Biology editors, an Academic Editor with relevant expertise, and by several independent reviewers. Please accept our apologies for the delayed decision, which was related to the recent holidays as well as a reviewers' need for extra time to complete their report.

In light of the reviews, which you will find at the end of this email, we would like to invite you to revise the work to thoroughly address the reviewers' reports.

As you will see below, the reviewers expressed interest in the topic and described the study as sophisticated. However, Reviewer 1 raised a concern about the effective stimulus concentration in this experimental context. Reviewer 2 raised concerns about some of the analytical choices and requested an additional control analysis. Reviewer 2's report also indicated that some claims were overstated, alternative interpretations should be discussed, and some areas require clarifications. In your revision, you fully address the reviewers' concerns.

Given the extent of revision needed, we cannot make a decision about publication until we have seen the revised manuscript and your response to the reviewers' comments. Your revised manuscript is likely to be sent for further evaluation by all or a subset of the reviewers.

**IMPORTANT - SUBMITTING YOUR REVISION**

*Re-submission Checklist*

*Published Peer Review*

*PLOS Data Policy*

*Blot and Gel Data Policy*

Sincerely,

Taylor

Taylor Hart, PhD,

Associate Editor

PLOS Biology

thart@plos.org

REVIEWS:

Reviewer #1: The manuscript by Norden et al., uses EEG/EBG recordings to analyze neural activity from the olfactory bulb and piriform cortex during odor delivery in humans. From these experiments, the researchers conclude that 1) oscillatory dynamics in both the olfactory bulb and piriform cortex primarily encode subjective perceived intensity, rather than actual concentration, 2) gamma-band activity in the olfactory bulb encoding perceived intensity is transmitted to the piriform cortex, and 3) the piriform cortex sends top-down beta band feedback to modulate OB activity via phase-amplitude coupling.

The topic is of interest as the neural mechanisms underlying intensity coding in olfactory research is poorly understood. Research in animal models have primarily focused on concentration, rather than intensity coding, given the difficulty of obtaining perceptual measures of intensity in non-human subjects. In contrast, the current study correlates neural activity with perceived intensity on a trial-by-trial basis, providing unique insight into the mechanisms of intensity coding.

I just have one major concern that relates to finding #1.

1.The authors find surprisingly little evidence of concentration-related modulation. This was somewhat shocking given the wealth of animal data supporting concentration-dependent coding in the olfactory bulb and piriform cortex. Thus, I am left wondering whether the authors can provide evidence that the vapor-phase concentration of these different liquid dilutions have 5-fold, 12-fold, and 14-fold differences between the low and high intensity stimuli. As the authors are no doubt aware, the relationship between liquid and vapor-phase concentrations are rarely linear at higher concenrations (Cometto-Muniz et al., 2003 and more). If these issues extend to the odorants/solvents used in this study, these positive deviations related to solvent-odorant interactions, could be skewing their data - thereby preventing an accurate test of concentration coding.

a.The authors state, "even when using an "effective concentration" metric (incorporating vapor pressure and dilution to linearize physical concentration across odorants), there was still no association between neural dynamics and concentration. These analyses assume ideal vapor-phase concentration conditions, which are very unlikely to be accurate. Thus, this is not an accurate test of effective concentration coding. However, to be fair, many animal studies also incorrectly assume ideal vapor-phase concentrations. Although their tested concentration ranges are often larger, which may help mitigate this potential issue?

b.I attempted to look for vapor-/liquid-phase equations for these odorants and solvents. Jennings et al., 2023 did provide data for 1-butanol in diethyl phthalate. Their data indicate a ~6.3 fold rather than 5-fold change for this closely related odorant. (0.8% - 1342*0.8^1 = 1073 and 5% - 1342*5^1 = 6710). However, they also mentioned that this relationship only holds for concentrations below 3.5% (and not the 5% used in this study). I don't know if the authors can find any additional sources of information.

c.A related concern stems from both the Jennings et al., 2023 and Commetto-Muniz et al., 2003 manuscripts. They observed that the relationship between liquid and vapor phase concentration can be "flatter" for longer chain odorants (ie exponents much lower than 1). If this is true for nonanone and undecanal, their vapor-phase concentration might be more "compressed" in comparison to the liquid concentration. This could theoretically skew the data and influence the statistical tests. However, this is just a supposition, I don't have any evidence for these particular odorants in diethyl phthalate.

Reviewer #2: This study investigates how physical concentration vs. perceived intensity influence scalp electroencephalographic (EEG) activity in response to different odorants. Using advanced statistical approaches, the authors analyzed time-frequency power spectra in both the olfactory bulb (OB) and piriform cortex (PC). Their findings indicate that neural activity in these regions is driven primarily by perceived odor intensity rather than by physical concentration. More specifically, the authors report a sequence of gamma-band activity followed by beta-band activity within the first 2 seconds after odorant onset. They interpret this temporal pattern as reflecting an early bottom-up transmission of perceived intensity from OB to PC, followed by top-down modulation from PC back to OB via phase-amplitude coupling and transient beta bursts. This proposed dynamic is suggested to represent a neurocognitive mechanism underlying odor intensity coding. Overall, the manuscript is well written and provides a sophisticated exploration of odor perception using human scalp EEG, contributing valuable findings to an area that remains relatively understudied. However, as outlined below, I have two major concerns regarding data analysis and interpretative overreach that should be addressed before I can recommend the paper for publication.

My first major concern relates to the analysis pipeline. More comprehensive analyses are needed to substantiate the reported results and to determine whether the findings truly capture the full picture. I outline below several specific points supporting this concern.

1/ Lines 102-104, the authors state "Although the significant cluster began earlier in the OB, both regions showed intermittent beta-band peaks as early as ~200 ms after odor onset; these early peaks did not survive cluster correction." Then, on lines 112-115, they write "As in the beta band, early OB gamma effects (~200 ms; 70-80 Hz) were present but did not survive cluster correction. However, restricting analysis to an a priori window (100-500 ms) did yield a cluster-corrected effect, consistent with the hypothesis that early OB gamma carries information about perceived intensity." How was this a priori window defined? And why was this restricted window applied only to gamma-band activity related to perceived intensity in the OB? Visual inspection of Figure 2 suggests early activity within this window not only in the OB but also in the PC, and for both odorant concentration and perceived intensity in the latter, with no obvious differences between them. These early responses may have been masked by the long-lasting clusters of late activity in the full 2-second cluster-based analysis. There is no clear rationale for restricting the analysis to an early window only for OB gamma activity associated with perceived intensity, particularly in light of the results from the later decoding analysis.

2/ To ensure that the observed effects truly reflect neural processes specific to odor perception, the authors should include analyses from one or more control brain regions to demonstrate the absence of similar effects elsewhere. This would help confirm that the reported findings are not merely generalizable across non-target regions.

3/ The authors should make the raw data publicly available in a dedicated repository. In addition, more detailed data should be presented either in the main manuscript or the supplementary materials:

-Please provide the power spectra for each perceived intensity and each concentration rather than only statistical maps of the contrasts between conditions.

-Figures S1 and S2 should be shown separately for the high and low concentrations to depict the relationship between concentration and perceived intensity.

My second major concern is that the authors make strong claims without sufficiently considering alternative interpretations or acknowledging the limitations of the evidence provided. I outline below several points that pertain to this concern.

1/ Throughout the manuscript, the authors apply sophisticated statistical tools to localize EEG sources, infer the direction of information flow between the OB and PC, etc. However, these methods have intrinsic limitations to infer the neural mechanisms they intend to reveal (see e.g., Stokes & Purdon, 2017, PNAS, for the problems encountered in Granger causality analysis applied to neuroscience). Yet, the authors state lines 297-298 "These findings establish a mechanistic and temporally resolved model of how subjective olfactory intensity is constructed in the human brain". Relying solely on statistical modeling of scalp EEG to infer causal neurocognitive architecture is questionable, and this limitation should be clearly acknowledged.

2/ In addition, several neurocognitive processes proposed by the authors to interpret the data are debated in the literature. For instance, the assumption that increases in power within a given frequency band reflect functional endogenous oscillatory activity remains disputed (e.g., Doelling & Assano, 2021, Plos Biol) and should be made with caution given the well-known "Fourier fallacy" (Jasper, 1948, Science). Similarly, the interpretation that high-frequency activity indexes local processing and low-frequency activity broader networks has been challenged by recent intracranial EEG findings (Jacques et al., 2022, eLife). Therefore, the manuscript would benefit from a more nuanced discussion of the neural interpretations.

3/ One clear example of an alternative interpretation relates to the claim on lines 250-251: "revealing that perception is manifested remarkably early in the human olfactory hierarchy". The temporal dynamics reported in the study, within a 2-s window, are in fact relatively slow compared to what is known about the speed of sensory perception in general, and olfactory perception in particular. In rodents, odor perception can occur within approximately 500 ms (e.g., Bolding & Franks, 2017, eLife). Although human processing may be slower due to larger brain size, behavioral latency studies suggest that 1 s is sufficient for many olfactory processes (reviewed in Olofsson, 2014, Front Psychol), and this same review notes that "major olfactory decisions may be confined to a rapidly unfolding cascade of inter-connected processing steps within a 500 ms interval" (p. 6).

Accordingly, what exactly does the late beta activity observed from 1 s onward represent? Its latency suggests that it may be more related to the task structure, namely waiting for 2 seconds before providing an intensity judgment, rather than odor perception per se. Classic work by Walter (1964, Nature) showed that sensorimotor associations and expectancy elicit a slow negative potential, the contingent negative variation (CNV), and later studies have linked CNV amplitude and latency to beta‑band power in time-frequency analyses (e.g., Bickel et al., 2012, NeuroImage). Thus, the late beta activity reported here may disappear under passive sniffing conditions without an upcoming explicit task. As noted above, including analyses of a control brain region not expected to show odor‑related responses would help clarify this issue, and this possibility should be discussed.

Minor points

I do not understand whether "perceived intensity" is treated as a continuous variable or as a dichotomized variable (i.e., median-split) in the first analysis using a linear mixed-effects model. This should be made explicit. If the authors indeed used a median split, they should additionally provide an analysis treating perceived intensity as a continuous predictor to verify that the results hold, given the well-documented statistical drawbacks of median-split procedures (e.g., MacCallum et al., 2002, Psych Methods).

Regarding the interpretation of differences between OB and PC, the authors write on lines 104-107: "Around 1 s into the trial, the association was stronger in the PC (t = 4.65, p = 3.36×10⁻⁶, CI = [.0757, .186]) than in the OB (t = 4.08, p = 4.49×10⁻⁵, CI = [.0726, .210]), suggesting that the PC holds a richer representation of perceived intensity at this time point." Similarly, on lines 110-112, they state: "Associations were comparable in magnitude for PC (t = 2.37, p = .022, CI = [.020, .064]) and OB (t = 2.41, p = .02, CI = [.021, .061])." However, these statements appear to rely solely on descriptive differences in t-values. To substantiate claims about differential sensitivity between OB and PC, the authors should directly compare power spectra between regions.

---

## [Decision Letter · Decision Letter 2]

1 Apr 2026

Dear Dr Nordén,

Thank you for your patience while we considered your revised manuscript "Olfactory bulb-cortex oscillations encode perceived odor intensity rather than concentration" for publication as a Research Article at PLOS Biology. Your revised study has been evaluated by the PLOS Biology editors, the Academic Editor, and the original reviewers.

In light of the reviews, which you will find at the end of this email, we would like to invite you to revise the work to thoroughly address the reviewers' reports.

As you'll see, while Reviewer 1 was satisfied, Reviewer 2 has some remaining technical and interpretational concerns requiring a thorough response. We therefore invite you to perform another Major Revision to address these comments.

Given the extent of revision needed, we cannot make a decision about publication until we have seen the revised manuscript and your response to the reviewers' comments. Your revised manuscript is likely to be sent for further evaluation by all or a subset of the reviewers.

**IMPORTANT - SUBMITTING YOUR REVISION**

*Re-submission Checklist*

*Published Peer Review*

*PLOS Data Policy*

*Blot and Gel Data Policy*

Sincerely,

Taylor

Taylor Hart, PhD,

Associate Editor

PLOS Biology

thart@plos.org

REVIEWS:

Reviewer #1: The authors have addressed all of my concerns.

Reviewer #2: The authors have adequately addressed several of the issues I raised, and the manuscript has substantially improved. However, I still have a number of concerns that should be addressed before I can recommend this paper for publication.

1/ In the previous review, I emphasized a lack of justification for restricting the cluster-based analysis to an early time window only for OB gamma activity related to perceived intensity. In response, the authors simply removed the analysis. However, as I already noted, Figure 2 shows early neural activity within the early 100-500 ms window in the beta band not only in the OB but also in the PC, and for both odorant concentration and perceived intensity in the PC. I also pointed out that early effects may be obscured by the presence of late and long-lasting significant clusters. Hence, my comment was rather to encourage the authors to examine early activity across all conditions and all frequency bands (see also my final comment below).

2/ I appreciate the newly provided supplementary figures showing the raw power spectra for odor concentration and perceived intensity in both the OB and PC. Surprisingly, these figures suggest a strong effect of odor concentration on late (1-2 s) gamma activity, particularly in the OB, while the effect of perceived intensity does not visually appear markedly different from that of concentration. While visual inspection alone is not sufficient to draw conclusions, it is nonetheless puzzling that the pronounced dissociation between the effects of odor concentration and perceived intensity reported in the statistical analysis is not more clearly reflected in the raw data. Could the authors elaborate on this discrepancy?

3/ Previously, I proposed that the late beta activity could be related to task structure rather than odor perception per se because the requirement to wait for 2 seconds before reporting an intensity judgment may elicit expectancy processes measurable in beta-band activity. I therefore suggested that the late effects described in the manuscript might not appear in a passive sniffing paradigm. The authors acknowledged in their reply that passive tasks could be valuable but did not mention it in the manuscript. Rather, based on additional analyses showing that the effects are confined to the olfactory system, they have added a short rebuttal arguing that this specificity rules out a global, unspecific anticipatory mechanism.

While I agree that the new analyses effectively demonstrate that the effects are specific to the olfactory regions, they do not dismiss the involvement of expectancy-related processes. These processes are not necessarily unspecific. Previous studies, including the study I previously cited (Bickel et al., 2012, NeuroImage), show that late expectancy-related neural activity can covary with stimulus content when the content is relevant for the upcoming behavioral decision. This is directly analogous to the present study, in which perceived intensity determines the behavioral response. Therefore, the argument that expectancy processes would necessarily be global or unspecific is not convincing.

In addition, as I previously mentioned, and as reviewed by Olofsson (2014, Front Psychol), odor processing proceeds rapidly, with high-level perceptual and semantic components unfolding well within the first second following odor onset. Because this time course is based on behavioral reaction times, which include motor components, the underlying neural processes are likely to occur even earlier than 1 second. These points reinforce the importance of examining the early window more precisely (see my first comment) and the manuscript should explicitly discuss this limitation, acknowledge that late activity may conflate perceptual and task-related components, and note that passive paradigms would be beneficial in future studies.

---

## [Decision Letter · Decision Letter 3]

24 Apr 2026

Dear Dr Nordén,

Thank you for your patience while we considered your revised manuscript "Olfactory bulb-cortex oscillations encode perceived odor intensity rather than concentration" for publication as a Research Article at PLOS Biology. This revised version of your manuscript has been evaluated by the PLOS Biology editors, the Academic Editor, and the remaining original reviewer.

Based on the reviews, we are likely to accept this manuscript for publication. Please also make sure to address the following data and other policy-related requests.

IMPORTANT: Please ensure that your next revision addresses all of the following points:

**Financial disclosure statement:

Please add links to the funding agencies in the Financial Disclosure statement in the manuscript details.

**Ethics:

Please move ethical and consent-related information to a new, first sub-heading of the Materials and Methods section, titled "Ethics statement".

**Data and Code:

Thank you for indicating that all anonymized data and scripts required to reproduce the results are available through OSF. However, we could not access these items through the provided link ( "https://osf.io/t7hj2/" ), which redirected us to the OSF homepage. Please check that the link is correct (and provide a DOI, if possible) and that the items are publicly available so that we can see them prior to formally accepting the paper.

Please also note that we specifically require that the numerical data underlying some of the figure panels be made available. This applies to the following panels:

1CD

S9A

Regarding code, please also note that journal policy requires that all custom code and scripts be made permanently and publicly available.

Please also cite the location of the data clearly in all relevant main and supplementary Figure legends, e.g. “The data underlying this Figure can be found in S1 Data” or “The data underlying this Figure can be found in https://doi.org/10.5281/zenodo.XXXXX”

We expect to receive your revised manuscript within two weeks.

*Published Peer Review History*

*Press*

Sincerely,

Taylor

Taylor Hart, PhD,

Associate Editor

thart@plos.org

PLOS Biology

REVIEW

Reviewer #2: [This reviewer recommended to Accept the manuscript without providing further comments to the author]

---

## [Editor Report · Decision Letter 4]

5 May 2026

Dear Dr Nordén,

Thank you for the submission of your revised Research Article "Olfactory bulb-cortex oscillations encode perceived odor intensity rather than concentration" for publication in PLOS Biology. On behalf of my colleagues and the Academic Editor, Thorsten Kahnt, I am pleased to say that we can in principle accept your manuscript for publication, provided you address any remaining formatting and reporting issues. These will be detailed in an email you should receive within 2-3 business days from our colleagues in the journal operations team; no action is required from you until then. Please note that we will not be able to formally accept your manuscript and schedule it for publication until you have completed any requested changes.

PRESS

Sincerely,

Taylor

Taylor Hart, PhD,

Associate Editor

PLOS Biology

thart@plos.org